# Are E-cigarettes associated with postpartum return to smoking? Secondary analyses of a UK pregnancy longitudinal cohort

Sophie Orton [1], Lauren Taylor,[1] Libby Laing,[1] Sarah Lewis,[1] Michael Ussher,[2,3] Tim Coleman,[1] Sue Cooper [1]

[1]School of Medicine, University of Nottingham, Nottingham, UK
[2]Division of Population Health Sciences and Education, St George's University of London, London, UK
[3]Institute for Social Marketing and Health, University of Stirling, Stirling, UK

**Correspondence to**
Sophie Orton;
sophie.orton@nottingham.ac.uk

## ABSTRACT

**Objectives** Postpartum return to smoking (PPRS) is an important public health problem. E-cigarette (EC) use has increased in recent years, and in a contemporary UK pregnancy cohort, we investigated factors, including ECs use, associated with PPRS.

**Design** Secondary analyses of a longitudinal cohort survey with questionnaires at baseline (8–26 weeks' gestation), late pregnancy (34–36 weeks) and 3 months after delivery.

**Setting** 17 hospitals in England and Scotland in 2017.

**Participants** The cohort recruited 750 women who were current or recent ex-smokers and/or EC users. A subgroup of women reported being abstinent from smoking in late pregnancy (n=162, 21.6%), and of these 137 (84.6%) completed the postpartum questionnaire and were included in analyses.

**Outcome measures** Demographics, smoking behaviours and beliefs, views and experience of ECs and infant feeding.

**Results** 35.8% (95% CI 28% to 44%) of women reported PPRS. EC use in pregnancy (adjusted OR 0.34, 95% CI 0.13 to 0.85) and breast feeding (adjusted OR 0.06, 95% CI 0.02 to 0.24) were inversely associated with PPRS, while household member smoking at 3 months post partum was positively associated with PPRS (adjusted OR 11.1, 95% CI 2.47 to 50.2).

**Conclusion** EC use in pregnancy could influence PPRS. Further research is needed to confirm this and investigate whether ECs could be used to prevent PPRS.

## INTRODUCTION

Helping pregnant women quit smoking and remain abstinent in the long term is an important public health issue. In 2019/2020, 12.1% of women in England were smoking in early pregnancy,[1] however, around half attempt cessation after conception.[2] Unfortunately, relapse is common, with up to 75% returning to smoking within 12 months of giving birth.[3–5] Reducing maternal postpartum return to smoking (PPRS) would have substantial health benefits for both mother and baby;[6] women would reduce

### Strengths and limitations of this study

► This is the first study, to our knowledge, to explore use of E-cigarette (EC) in pregnancy and post partum, among other potential risk factors, for return to smoking postpartum.
► The cohort collected longitudinal, prospective data on women's smoking behaviour during pregnancy and postpartum, reducing recall error.
► The cohort is likely to be broadly representative as the demographic profile of smokers is similar to other UK pregnancy cohorts.
► The primary study was not designed to answer the research question; consequently, relatively small numbers of women reported returning to smoking post partum and using ECs.
► Smoking and EC use were self-reported, which could lead to under-reporting due to social stigma; however, the surveys were completed anonymously which may have minimised this issue.

their own risks of smoking-related illness, reduce ill health and mortality associated with secondhand smoke exposure among their children,[7 8] and reduce the likelihood of their children becoming smokers themselves in later life.[9]

There are currently no effective interventions for reducing PPRS.[10] Understanding the factors that are associated with PPRS is essential for the development of targeted and evidence-based interventions. A systematic review[11] observed that women that returned to smoking postpartum tended to be less well educated, younger, multiparous, living with a partner or household member who smoked, experiencing higher stress, depression or anxiety, not breast feeding, intending to quit only for pregnancy and having low confidence in remaining quit after giving birth.

Since this review, E-cigarettes (ECs) have become increasingly popular.[12] ECs have potential for public health benefit as they

do not involve tobacco combustion, which is the main source of harm from conventional cigarettes. In non-pregnant smokers, there is moderate certainty that ECs containing nicotine improve cessation rates, compared with non-nicotine vaping products or nicotine replacement therapy (NRT).[13] In 2021, ECs were used in 29% of UK quit attempts[12] and around 5% of UK women report using ECs in pregnancy.[14] There is also evidence from qualitative studies that some women are using ECs to avoid returning to smoking after having their baby.[15] Hence, using a contemporary UK pregnancy cohort, we investigated factors associated with PPRS, including EC use in pregnancy and afterwards.[11]

## METHODS

This paper follows the Strengthening the Reporting of Observational Studies in Epidemiology reporting guidelines.[16] We conducted secondary analyses on data collected in a UK longitudinal pregnancy cohort designed to explore the use of and attitudes towards ECs during pregnancy and postpartum. Full methods and cohort characteristics are presented elsewhere.[14] In brief, the cohort recruited women from 17 hospital antenatal clinics from June to November 2017. Hospitals had varying smoking in pregnancy rates and were from a range of geographical locations in England and Scotland. Screening surveys were systematically handed out to all pregnant women attending selected antenatal clinics. Women were eligible to join the cohort if they self-reported being either current or recent ex-smokers (quit smoking in the 3 months prior to pregnancy) and/or current EC users. Data were collected at 8–26 weeks gestation (baseline), 34–36 weeks gestation (follow-up 1, FU1) and 12 weeks post partum (follow-up 2, FU2). Baseline questionnaires asked about maternal demographics (age, education, ethnicity), parity, smoking behaviours and beliefs, and use of ECs. FU1 and FU2 questionnaires were similar to baseline but also asked about plans for infant feeding, how baby was being fed and smoking/EC use post partum.

### Patient and public involvement

Patient and public involvement colleagues contributed to the design of the study, including providing feedback on the importance of the research question, designing study materials and advising on study recruitment.

### Selection of respondents for inclusion in analyses

Women who reported having quit smoking either in the 3 months prior to pregnancy, or during pregnancy, and were still abstinent by FU1 (34–36 weeks gestation) were included in the analysis.

### Dependant and explanatory variables

The primary outcome measure was return to smoking at 3 months postpartum using participant's responses to the question 'which statement best describes your smoking

right now?'. Response options were 'I do not smoke at all' 'I smoke occasionally, but not every day', 'I smoke every day, but less than when I was pregnant', 'I smoke every day, about the same as when I was pregnant' and 'I smoke every day and tend to smoke more than when I was pregnant'. Women who reported smoking at least occasionally were considered to have returned to smoking.

The explanatory variables measured at baseline/during pregnancy were: baseline smoking status (current smoker or recent ex-smoker), education ('no qualifications' or 'general certificate of secondary education or above', using highest educational qualification), maternal age, previous pregnancies (no previous pregnancy' or 'previous pregnancy'), baseline Heaviness of Smoking Index (HSI) ('low', 'moderate' or 'high'),[17] NRT use in pregnancy ('not used NRT' or 'used NRT'), confidence in staying smoke free ('not at all—a little' or 'moderately—extremely'), whether they had accessed stop smoking support during pregnancy ('not accessed" or 'accessed'), and EC use during pregnancy ('never used EC in pregnancy' or 'used EC at some point during pregnancy'). Postpartum explanatory variables were breast feeding ('baby no longer/never breastfed' or 'baby still breast feeding at 12 weeks post partum'), household member smoking ('no household member smoking' or 'household member smokes' and EC use ('never used EC post partum' or 'used EC at some point postpartum').

### Analysis

Analyses were conducted using Stata V.13.1.[18] We handled missing data using available case analysis. Baseline demographic characteristics were analysed using descriptive statistics. Prevalence of PPRS was calculated for those who responded at FU2.

An exploratory logistic regression was conducted to identify factors associated with PPRS. Collinearity between variables was assessed using correlation analysis and no significant correlations were identified. HSI was excluded from logistic regression because all those in the PPRS group had low HSI. We used an iterative approach to determine which variables best predicted PRRS. Variables were selected based on those identified to be associated with PPRS in previous literature,[11] and those of theoretical interest (eg, EC use). We first determined which of these variables were significantly associated (p<0.05) with PPRS in univariate logistic regression. These variables were then included together in a multivariate logistic regression model. Variables that were not significant were removed from the model, so that only significant variables (p<0.05) were retained. Variables which had been excluded, including those that were not significant in univariate analysis, were added into the model consecutively to assess whether they became significant and retained if so. Effects were expressed as ORs and 95% CIs, and likelihood ratio test p values (p<0.05) are presented.

**Table 1** Baseline questionnaire characteristics of those participants who were non-smokers in late pregnancy and a comparison between those who did and did not respond to the postpartum survey

| Characteristics at baseline (8–26 weeks gestation) | Women abstinent from smoking in late pregnancy, N (%), N=162 | Responded to postpartum questionnaire, N (%), N=137 | Did not respond to postpartum questionnaire, N (%) N=25 |
|---|---|---|---|
| **Smoking status** | | | |
| Recent ex-smoker | 133 (82.1) | 116 (84.7) | 17 (68.0) |
| Current smoker | 29 (17.9) | 21 (15.3) | 8 (32.0)* |
| **HSI score** | | | |
| Low addiction | 28 (82.4) | 22 (84.6) | 6 (75.0) |
| Moderate addiction | 6 (17.7) | 4 (15.4) | 2 (25.0) |
| High addiction | 0 (0.0) | 0 (0.0) | 0 (0.0) |
| **Education** | | | |
| No qualifications | 11 (6.8) | 7 (5.15) | 4 (16.0) |
| GCSE or above (including other) | 150 (93.2) | 129 (94.9) | 21 (84.0)* |
| **Maternal age (mean, SD)** | 26.9 (5.6) | 27.1 (5.3) | 25.9 (7.0) |
| **Ethnicity** | | | |
| White British | 139 (86.3) | 118 (86.8) | 21 (84.0) |
| Other ethnicity | 22 (13.7) | 18 (13.2) | 4 (16.0) |
| **Previous pregnancy** | | | |
| No previous pregnancy | 73 (45.6) | 64 (47.1) | 9 (37.5) |
| Previous pregnancy | 87 (54.4) | 72 (52.9) | 15 (62.5) |
| **Confidence in staying smoke free** | | | |
| Not at all-a little | 20 (12.7) | 18 (13.5) | 2 (8.33) |
| Moderately-extremely | 137 (87.3) | 115 (86.5) | 22 (91.7) |
| **EC use** | | | |
| Not using | 152 (93.8) | 129 (94.2) | 23 (92.0) |
| Using | 10 (6.2) | 8 (5.8) | 2 (8.0) |

Percentages calculated from the total number of responders for each variable as not all questions had complete data.
*P<0.05.
EC, e-cigarette; GCSE, general certificate of education; HSI, Heaviness of Smoking Index.

## RESULTS

### Response rates

The total cohort included 750 women, and 162 (21.6%) were eligible and included in analyses. From these eligible women, response rate for the FU2 (post partum) survey was 84.6% (137/162) (online supplemental figure S1).

### Cohort characteristics

Table 1 presents the baseline characteristics of women who were abstinent from smoking in late pregnancy (ie, at FU1) and of respondents and non-respondents to the postpartum questionnaire (FU2). At baseline, 17.9% smoked, 82.4% had low addiction levels and 93.8% were not using ECs. 29.6% used an EC at some point in pregnancy, and 19.9% used an EC at some point in the 12 weeks since having their baby. Those who did not respond to FU2 were more likely to be a current smoker at baseline and have a lower educational level.

### Prevalence of PPRS

35.8% (n=49, 95% CI 28% to 44%) of women reported PPRS.

In univariate analysis (table 2), breast feeding at 12 weeks post partum was inversely associated with PPRS, while having a household member who was smoking post partum was positively associated with PPRS. PPRS was also less common in those who used EC in pregnancy but not significantly so (p=0.1).

In multiple logistic regression, the significant independent predictors of PPRS were breast feeding, use of ECs during pregnancy and household member smoking postpartum. The use of ECs during pregnancy (adjusted OR 0.34, 95% CI 0.13 to 0.85) and breast feeding (adjusted OR 0.06, 95% CI 0.02 to 0.24) were associated with lower risk of PPRS and a household member smoking postpartum with higher risk of PPRS (adjusted OR 11.1, 95% CI 2.47 to 50.2).

**Table 2** Univariate and multiple logistic regression showing associations with postpartum return to smoking

| Characteristics | All (women who quit smoking in pregnancy and completed FU2) (N=137), N | % Returning to smoking postpartum (N=49), N (row %) | Crude OR (95% CI) | Adjusted OR (95% CI) N=136 |
|---|---|---|---|---|
| **Baseline/pregnancy variables** | | | | |
| Baseline smoking status | | | | |
| Current smoker | 21 | 7 (33.3) | 0.88 (0.33 to 2.35) | |
| Ex-smoker | 116 | 42 (36.2) | Reference | |
| Education | | | | |
| No qualifications | 7 | 2 (28.6) | Reference | |
| GCSEs or above | 129 | 46 (35.7) | 1.39 (0.26 to 7.43) | |
| Maternal age | | | | |
| Continuous—mean (SD) | 27.1 | 26.7 (4.8) | 0.98 (0.92 to 1.05) | |
| Previous pregnancy | | | | |
| No previous pregnancy | 64 | 25 (39.1) | Reference | |
| Previous pregnancy | 72 | 23 (31.9) | 0.73 (0.36 to 1.48) | |
| Confidence in staying smoke free (baseline) | | | | |
| Not at all-a little | 18 | 8 (44.4) | Reference | |
| Moderately-extremely | 115 | 38 (33.0) | 0.62 (0.22 to 1.69) | |
| NRT use in pregnancy† | | | | |
| Not used NRT | 129 | 47 (36.4) | Reference | |
| Used NRT | 8 | 2 (25.0) | 0.58 (0.11 to 3.00) | |
| Accessed support from SSS in pregnancy† | | | | |
| Not accessed support | 128 | 47 (36.7) | Reference | |
| Accessed support | 9 | 2 (22.2) | 0.49 (0.10 to 2.47) | |
| EC use in pregnancy†‡ | | | | |
| Never used EC in pregnancy | 95 | 38 (40.0) | Reference | Reference |
| Used EC at some point during pregnancy | 42 | 11 (26.2) | 0.53 (0.24 to 1.19) | 0.34 (0.13–0.85)* |
| **Post partum** | | | | |
| Breast feeding | | | | |
| Baby no longer/never breastfed | 90 | 43 (47.8) | Reference | Reference |
| Baby still breastfeeding at 12 weeks post partum | 46 | 5 (10.9) | 0.13 (0.05 to 0.37)** | 0.06 (0.02–0.24)** |
| Household member smoking postpartum | | | | |
| No household member smoking | 120 | 39 (32.5) | Reference | |
| Household member smokes | 17 | 10 (58.8) | 2.97 (1.05 to 8.38)* | 11.1 (2.47–50.2)** |
| EC use postpartum‡ | | | | |
| Never used EC postpartum | 109 | 38 (34.9) | Reference | |
| Used EC at some point postpartum | 27 | 10 (37.0) | 1.10 (0.46 to 2.64) | |

Variables that do not total 137 are missing.
*P<0.05, **p<0.001.
†Data collated from baseline and FU1.
‡No correlation between EC use in pregnancy and EC use postpartum detected.
EC, electronic cigarette; FU, follow-up; GCSE, general certificate of secondary education; NRT, nicotine replacement therapy; SSS, Stop Smoking Services.

## EC use

Forty-two (30.7%) reported using an EC at some point in pregnancy, and 27 (19.9%) reported using an EC at some point postpartum. 20 (48.8%) of those who used an EC in pregnancy also used an EC post partum.

## DISCUSSION
### Key findings

This is the first study, to our knowledge, to explore use of ECs in pregnancy and postpartum, among other potential odds of PPRS. Using ECs in pregnancy, breast feeding and not having a household member who smokes are associated with a lower likelihood of returning to smoking post partum.

### Strengths and weaknesses

The cohort collected longitudinal, prospective data on women's smoking behaviour during pregnancy and post partum, reducing recall error. The cohort is likely to be broadly representative as the demographic profile of smokers is similar to other UK pregnancy cohorts,[14] and the characteristics of the subsample included in our analyses reflect those known to be associated with smoking cessation in pregnancy.[19]

A limitation is that the primary study was not designed to answer the research question; consequently, relatively small numbers of women reported returning to smoking postpartum and using ECs. This could limit the reliability of our findings; however, in previous research, not breast feeding and household member smoking are commonly associated with PPRS,[11] suggesting our sample size was sufficient to detect some potentially relevant associations, but it is possible that more subtle ones were missed. We measured HSI, which was low in the majority of women; PPRS only occurred in those with low HSI so we could not look at its independent effect. Reliance on self-reported smoking and EC use may lead to under-reporting, particularly due to social stigma.[20 21] However, the surveys were anonymous, baseline surveys were completed discreetly in clinic and follow-ups were conducted in the privacy of home. As smoking prevalence varies internationally,[22] the prevalence of EC use is also likely to vary, and these findings may only be generalisable to countries like the UK, where ECs are encouraged for smoking cessation. This cohort study was conducted in 2017–2018, since which time EC awareness, marketing and health advice may have changed. However, as there are no similar published analyses on this topic our findings are an important contribution to the scientific literature.

### Findings in context of previous literature

Our findings are consistent with previous studies in which less PPRS was observed among mothers who breastfed and did not live with smokers.[11] To our knowledge, this is the first time that an association between EC use in pregnancy and PPRS has been observed and the reason for this inverse relationship is not immediately clear.

Previous literature suggests that EC use, smoking and quitting may influence each other. Qualitative research has reported that one of the primary motivations for using ECs in pregnancy and postpartum is to quit smoking or prevent returning to smoking.[23] In non-pregnant populations, a meta-analysis of three studies found that risk of relapse was higher among former smokers who used ECs (relative risk 1.38, 95% CI 1.11 to 1.65), although no relationship was observed between relapse and frequency of EC use.[24] Conversely, qualitative research with non-pregnant former smokers observes that using ECs can support smoking relapse prevention by offering a long-term alternative to tobacco, and can prevent a smoking lapse becoming a full relapse.[25] Among ECs users in this cohort, the most commonly reported reasons for using ECs both in pregnancy and post partum were to quit smoking and avoid returning to smoking.[14] We would, therefore, have expected to also see a similar relationship between postpartum EC use and PPRS. It may be that our sample size was too small to detect this association, or it is possible that other, non-measured confounders are responsible for our findings. Future research is needed to confirm the relationship between EC use and reduced PPRS and establish whether this association could be causal.

## CONCLUSION

We believe this is the first demonstration that EC use in pregnancy could influence PPRS. Further research is needed to confirm this finding and investigate the possibility that ECs could be used to prevent returning to smoking in the post partum.

**Contributors** Conceptualisation: SO, TC, MU, SC and SL; methodology, SO, TC, MU, SC and SL; formal analysis, LT, LL and SO; writing—original draft preparation, SO, LT and LL; writing—review and editing, SO, LT, LL, TC, MU, SC and SL; supervision, SO, TC, SC, SL and MU; project administration, SO, LT and LL; funding acquisition, SO, TC, SC, SL and MU. Guarantor, SO. All authors read and agreed to the published version of the manuscript.

**Funding** This study/project is funded by the National Institute for Health Research (NIHR) School for Primary Care Research (NIHR SPCR-2016-102). TC is an NIHR Senior Investigator. The original cohort study was funded by Cancer Research UK, Tobacco Advisory Group Project (grant number C53479/A22733).

**Disclaimer** The views expressed are those of the author(s) and not necessarily those of the NIHR or the Department of Health and Social Care.

**Competing interests** None declared.

**Patient and public involvement** Patients and/or the public were involved in the design, or conduct, or reporting, or dissemination plans of this research. Refer to the Methods section for further details.

**Patient consent for publication** Consent obtained directly from patient(s).

**Ethics approval** Ethical approval was granted by the South West Frenchay Research Ethics Committee (IRAS project ID 222886). Participants gave informed consent to participate in the study before taking part.

**Provenance and peer review** Not commissioned; externally peer reviewed.

**Data availability statement** Data are available on reasonable request. All data sharing requests can be made to SC (principal investigator to the original cohort).

**ORCID iDs**
Sophie Orton http://orcid.org/0000-0002-8577-216X
Sue Cooper http://orcid.org/0000-0002-1994-6395

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
