## [Reviewer comments · BMJ Open]

ARTICLE DETAILS

TITLE (PROVISIONAL)	ARE E-CIGARETTES ASSOCIATED WITH POSTPARTUM RETURN TO SMOKING? SECONDARY ANALYSES OF A UK PREGNANCY LONGITUDINAL COHORT
AUTHORS	Orton, Sophie; Taylor, Lauren; Laing, Libby; Lewis, Sarah; Ussher, Michael; Coleman, Tim; Cooper, Sue

VERSION 1 – REVIEW

REVIEWER	Adam Cole University of Ontario Institute of Technology
REVIEW RETURNED	04-Feb-2022

GENERAL COMMENTS	Postpartum return to smoking (PPRS) is an important public health issue as it affects the health of the mother and child. While there is a systematic review the identified factors associated with PPRS, it included studies and was completed before e-cigarettes were widely available and popular. This study used secondary data from a cohort in the UK to identify factors associated with PPRS, including associations with e-cigarette use during and post pregnancy. The results indicate that e-cigarette use in pregnancy led to lower odds of PPRS. There is relevant background information presented in the Introduction and the presentation of the Methods and Results is generally clear. I have a few comments to clarify some points in the Methods and Results. -Page 6 line 51: How much missing data was present? Of the 137 participants included in the analyses, it's unclear how many would be excluded from the regression models due to missing data. Given that those with missing data are excluded from the regression model, why not use a complete case analysis? -Page 6 Analysis: It's not clear which waves of data are being used in the analyses and how. Are all three waves of data used? Based on the description of the Methods, some questions were asked multiple times, but it's not clear which responses are being used in these analyses. This might be clarified through consistent use of timepoint language (i.e., baseline, FU1, FU2) throughout the manuscript. It doesn't appear as though changes in variables are being modeled. As described, it doesn't seem like this is a longitudinal data analysis, but rather a cross-sectional analysis using a longitudinal sample. -Page 7 line 12: How did the authors handle the multilevel nature of the data? Given the use of a longitudinal sample, FU1 and FU2 data are going to be highly correlated with baseline data. This may be particularly important for e-cigarette use outcomes. E-cigarette use at baseline would be correlated with e-cigarette use in pregnancy and postpartum (it is likely that those who use e-cigarettes at baseline continue to use them at follow-up waves). Multilevel regression modeling may be necessary.
--

	-Page 7 line 13: What was the rationale for this step-wise modelling approach rather than including/adjusting for all variables of interest? -Page 7 line 44: It appears as though many women in the sample used e-cigarettes either during or after pregnancy. Did the authors asked the women for their reason for using/trying e-cigarettes? Such data would relate to the author's Discussion (page 12 line 25). -Table 1 legend: What kind of test was used to calculate the p-values? This was not explained in the Analysis. -Page 10 line 3: the sentence wording should report the "odds of PPRS", not the risk given that the authors have calculated odds ratios. -Table 2: Column 2 was previously reported in Table 1, so I don't think it needs to be reported again here. It might be more relevant to show and compare the characteristics of those who did and did not return to smoking postpartum, as this would relate to the calculated odds ratios. -Page 12 line 15: given the findings that e-cigarette use lowered the likelihood of PPRS, it might be useful to report the percentage/proportion of participants who were e-cigarette users at baseline who continued using e-cigarettes at FU1 and FU2. These data could help to reinforce the hypothesis that pregnant women may use e-cigarettes to prevent relapse.
--	---

REVIEWER	Amy Broadfield University of Lincoln
REVIEW RETURNED	07-Feb-2022

GENERAL COMMENTS	Overall a very well-written paper that sheds light on a very interesting topic. I think there could be an addition of choice of theoretical framework when designing the survey tool.
--

VERSION 1 – AUTHOR RESPONSE

Reviewer: 1

We thank the reviewer for their interest in our study and suggestions to improve our paper. We have addressed their comments below:

-Page 6 line 51: How much missing data was present? Of the 137 participants included in the analyses, it's unclear how many would be excluded from the regression models due to missing data. Given that those with missing data are excluded from the regression model, why not use a complete case analysis?

We thank the reviewer for their comment. Although not explicitly reported, missing data from individual variables can be identified in tables 1 and 2 for those variables that do not total N=162/N=137 (highlighted in table footnotes). The final regression model included n=136, and we have now added this information into table 2 (column 5).

We agree with the reviewer that it is important to consider complete case analyses. We therefore conducted a sensitivity analysis on complete cases, with no substantial differences observed in our findings compared to our original available case analyses. Missing data therefore does not appear to have created biases that are sufficiently large enough to change our findings.

-Page 6 Analysis: It's not clear which waves of data are being used in the analyses and how. Are all three waves of data used? Based on the description of the Methods, some questions were asked

multiple times, but it's not clear which responses are being used in these analyses. This might be clarified through consistent use of timepoint language (i.e., baseline, FU1, FU2) throughout the manuscript. It doesn't appear as though changes in variables are being modeled. As described, it doesn't seem like this is a longitudinal data analysis, but rather a cross-sectional analysis using a longitudinal sample.

We thank the reviewer for raising this question and agree that it requires clarification. Some variables included in our analyses were based on reports from across pregnancy (table 2: NRT use in pregnancy, stop smoking support accessed in pregnancy, e-cigarette use in pregnancy), and so contain data collated from baseline and FU1. Other variables (table 2: baseline smoking status, education, maternal age, previous pregnancies, confidence in staying smokefree) were using baseline only data and are identified by '(baseline)' in table 2. Data from FU2 is clearly separated as a distinct section of table 2.

To aid clarification, we have gone through the paper and ensured that our description of baseline, FU1 and FU2 remains consistent throughout. We have added a footnote to table 2 to identify those variables containing data collated from baseline and FU1, and the order of variables presented in table 2 has been changed to make this clearer.

We would agree with the reviewer's description of our paper reporting cross-sectional analysis using a longitudinal sample as there are no 'repeated measures' included in our analysis (hence we did not use multilevel modelling, as discussed below).

-Page 7 line 12: How did the authors handle the multilevel nature of the data? Given the use of a longitudinal sample, FU1 and FU2 data are going to be highly correlated with baseline data. This may be particularly important for e-cigarette use outcomes. E-cigarette use at baseline would be correlated with e-cigarette use in pregnancy and postpartum (it is likely that those who use e-cigarettes at baseline continue to use them at follow-up waves). Multilevel regression modeling may be necessary.

We thank the reviewer for highlighting this important issue. We considered multilevel regression modelling, however the only repeated measure within our analyses was e-cigarette use; no correlation between e-cigarette use in pregnancy and e-cigarette use postpartum was detected (highlighted in footnote to table 2). Multilevel modelling was therefore not appropriate for our data.

-Page 7 line 13: What was the rationale for this step-wise modelling approach rather than including/adjusting for all variables of interest?

We thank the reviewer for raising this important question. We did not use traditional stepwise modelling (this approach has been widely criticised, e.g. *Smith G. Step away from stepwise. Journal of Big Data 2018;5(1):1-12.*). Instead, we used a more nuanced approach; we first identified all variables of interest based on previous literature (Orton S, Coleman T, Coleman-Haynes T, et al. Predictors of Postpartum Return to Smoking: A Systematic Review. *Nicotine Tob Res 2017;20(6):665-73. doi: 10.1093/ntr/ntx163*), which were then iteratively removed and re-added so that only significant variables were retained. This ensured that theoretical significance and statistical significance were both important considerations in the final model.

To aid clarification, the analysis section now states:

"We used an iterative approach to determine which variables best predicted PPRS. Variables were selected based on those identified to be associated with PPRS in previous literature,¹¹ and those of theoretical interest (e.g. EC use). We first determined which of these variables were significantly associated ($p < 0.05$) with PPRS in univariate logistic regression. These variables were then included together in a multivariate logistic regression model. Variables that were not significant were removed from the model, so that only significant variables ($p < 0.05$) were retained. Variables which had been excluded, including those that were not significant in univariate analysis, were added into the model consecutively to assess whether they became significant and retained if so."

-Page 7 line 44: It appears as though many women in the sample used e-cigarettes either during or after pregnancy. Did the authors ask the women for their reason for using/trying e-cigarettes? Such data would relate to the author's Discussion (page 12 line 25).

We thank the reviewer for their comment. Reasons for using e-cigarettes within this cohort were examined and reported elsewhere (Bowker K, Lewis S, Ussher M, et al. Smoking and vaping patterns during pregnancy and the postpartum: A longitudinal UK cohort survey. *Addict Behav* 2021;123:107050. doi: <https://doi.org/10.1016/j.addbeh.2021.107050>). The most commonly reported reasons for using e-cigarettes were to quit smoking and avoid smoking relapse. We agree that this would be useful to include in the discussion of our findings, and have added the following:

"Among ECs users in this cohort, the most commonly reported reasons for using ECs both in pregnancy and postpartum were to quit smoking and avoid returning to smoking¹⁴. We would therefore have expected to also see a similar relationship between postpartum EC use and postpartum return to smoking."

-Table 1 legend: What kind of test was used to calculate the p-values? This was not explained in the Analysis.

We thank the reviewer for highlighting this omission, and have included the following in the analysis section:

"Effects were expressed as odds ratios and 95% confidence intervals, and likelihood ratio test p-values ($p < .05$) are presented."

-Page 10 line 3: the sentence wording should report the "odds of PPRS", not the risk given that the authors have calculated odds ratios.

We thank the reviewer for this comment and have made the change as recommended.

-Table 2: Column 2 was previously reported in Table 1, so I don't think it needs to be reported again here. It might be more relevant to show and compare the characteristics of those who did and did not return to smoking postpartum, as this would relate to the calculated odds ratios.

We thank the reviewer for highlighting this issue. We have now amended table 2 so that column 2 presents data for the whole sample included in the univariate and multivariate analyses ($n=137$, previously it was $n=162$ for those who reported being abstinent in late pregnancy), and the corresponding column percentages reported in column 3. We have removed the percentages from column 2 to aid clarity. Univariate and multivariate findings (columns 4 and 5) in table 2 remain unchanged.

Whilst we acknowledge that there is some overlap in the data presented in tables 1 and 2, not all the data is duplicated and the tables are reporting different information (e.g. table 1 is looking at characteristics of responders and non-responders to FU2, table 2 is presenting univariate and multivariate analyses). We believe it is important to present participant characteristics alongside univariate and multivariate analyses, as we have done in table 2.

-Page 12 line 15: given the findings that e-cigarette use lowered the likelihood of PPRS, it might be useful to report the percentage/proportion of participants who were e-cigarette users at baseline who continued using e-cigarettes at FU1 and FU2. These data could help to reinforce the hypothesis that pregnant women may use e-cigarettes to prevent relapse.

We thank the reviewer for this comment.

We have added the following information to our results section:

“E-cigarette use

42 (30.7%) reported using an EC at some point in pregnancy, and 27 (19.9%) reported using an EC at some point postpartum. 20 (48.8%) of those who used an EC in pregnancy also used an EC postpartum.”

More detailed information on the patterns of smoking and e-cigarette use longitudinally across the cohort has been reported in detail elsewhere (Bowker K, Lewis S, Ussher M, et al. Smoking and vaping patterns during pregnancy and the postpartum: A longitudinal UK cohort survey. *Addict Behav* 2021;123:107050. doi: <https://doi.org/10.1016/j.addbeh.2021.107050>).

Reviewer: 2

Miss Amy Broadfield, University of Lincoln Comments to the Author:

Overall a very well-written paper that sheds light on a very interesting topic.

I think there could be an addition of choice of theoretical framework when designing the survey tool.

We thank the reviewer for their positive comments and taking the time to review our paper.

This paper presents secondary analyses of a longitudinal cohort survey, and so we were not able to input into the original survey design. We acknowledge this as a limitation in the discussion. The original survey has been described elsewhere and is referenced within the paper.

VERSION 2 – REVIEW

REVIEWER	Adam Cole University of Ontario Institute of Technology
REVIEW RETURNED	15-Mar-2022
GENERAL COMMENTS	The authors have sufficiently addressed my previous comments. I have no additional comments for the authors.